# First Observations of Buzzards (*Buteo*) as Definitive Hosts of *Sarcocystis* Parasites Forming Cysts in the Brain Tissues of Rodents in Lithuania

**DOI:** 10.3390/biology13040264

**Published:** 2024-04-16

**Authors:** Petras Prakas, Marius Jasiulionis, Tautvilė Šukytė, Evelina Juozaitytė-Ngugu, Vitalijus Stirkė, Linas Balčiauskas, Dalius Butkauskas

**Affiliations:** Nature Research Centre, Akademijos Str. 2, 08412 Vilnius, Lithuania; petras.prakas@gamtc.lt (P.P.); marius.jasiulionis@gamtc.lt (M.J.); tautvile.sukyte@gamtc.lt (T.Š.); evelina.ngugu@gamtc.lt (E.J.-N.); vitalijus.stirke@gamtc.lt (V.S.); dalius.butkauskas@gamtc.lt (D.B.)

**Keywords:** small mammals, *Sarcocystis*, brain tissues, *Buteo* hawks, life cycle, prevalence, genetic identification, *28S* rRNA, phylogeny

## Abstract

**Simple Summary:**

Some species of *Sarcocystis* parasites form cysts in the brains of small mammals. These parasites have been shown in laboratory experiments to be transmitted by *Buteo* buzzards. However, there is a lack of studies identifying these parasites in natural definitive hosts. In the current investigation, we examined brain tissues of small mammals and small intestines of two buzzard species collected in Lithuania for *Sarcocystis* spp. Species of *Sarcocystis* were confirmed using DNA sequence analysis. Of the eleven small mammal species inspected, only bank voles were infected with cysts of *Sarcocystis glareoli*. The prevalence of this parasite in the brain of vole hosts reached 9.1%. Based on genetic examination, half of the investigated common buzzards were positive for *S*. *glareoli*. Furthermore, two *Sarcocystis* species, including a putative new species, were detected in the small intestines of rough-legged buzzards. Our results indicate that *Buteo* buzzards play an important role in transmitting rarely studied *Sarcocystis* parasites forming cysts in the tissues of small mammals.

**Abstract:**

Representatives of the genus *Sarcocystis* are worldwide distributed apicomplexan parasites characterised by two-host prey-predator relationships. *Sarcocystis* spp. produce sarcocysts in the muscles and brains of intermediate hosts and develop sporocysts in the intestines of definitive hosts. Two species, *Sarcocystis glareoli* and *Sarcocystis microti*, previously assigned to the genus *Frenkelia*, form cysts in the brains of rodents and are transmitted through the common buzzard (*Buteo buteo*). In our study, brain samples of 694 small mammals caught in different regions of Lithuania were examined for *Sarcocystis* spp. Additionally, 10 *B. buteo* and two rough-legged buzzards (*Buteo lagopus*) were tested for sporocysts of the analysed parasites. *Sarcocystis* species were identified based on *28S* rRNA sequence comparison. Of the eleven species of small mammals tested, *Sarcocystis* parasites were observed only in the bank vole (*Clethrionomys glareolus*). Cysts of *S*. *glareoli* were detected in 34 out of 374 *C. glareolus* (9.1%, 95% CI = 6.4–12.5%). Molecular investigation showed the presence of only *S*. *glareoli* in the intestines of 50% of *B. buteo*. Furthermore, two species, *Sarcocystis* sp. Rod3 and *Sarcocystis* sp. Rod4, were confirmed in *B. lagopus*. Our results demonstrate the need for further studies on *Sarcocystis* cycling between rodents and birds.

## 1. Introduction

The genus *Sarcocystis* (Apicomplexa: Sarcocystidae) is a widespread group of parasites worldwide, comprising more than 200 species. These parasites infect reptiles, birds and mammals, including humans. Members of the genus *Sarcocystis* are characterised by a two-host life cycle. Typically, the intermediate host of the parasite is the prey, and the definitive host is the predator. Sarcocysts develop mainly in the muscles and central nervous system (CNS) of the intermediate host, while oocysts sporulate in the small intestine of the definitive host [1,2]. Some species of *Sarcocystis* are pathogenic, and the pathogenicity of the parasite is expressed in intermediate hosts [3].

Due to their widespread prevalence, small mammals are fundamental components in food webs, playing important roles in ecosystem functioning. They serve as the base of the food chain for many birds of prey [4,5]. Small mammals represent one of the animal groups where the highest number of *Sarcocystis* species has been identified. To date, over 45 different *Sarcocystis* species have been reported to use small mammals as intermediate hosts [1], and *S*. *attenuati* [6], *S*. *funereus* [7], *S*. *kani* [8], *S*. *muricoelognathis* [9], *S*. *myodes* [10], *S*. *ratti* [11] and *S*. *scandentiborneensis* [12] have been described in recent years. However, as surveys for *Sarcocystis* spp. in these hosts are fragmentary, the actual number of *Sarcocystis* species is likely much higher [8,13].

In previous taxonomical reviews, members of the family Sarcocystidae were traditionally categorized into two groups. The genera *Sarcocystis* and *Frenkelia* were classified under the subfamily Sarcocystinae, while other genera such as *Besnoitia*, *Hammondia*, *Neospora*, *Toxoplasma*, and others were attributed to subfamily Toxoplasamatinae. Within the genus *Frenkelia*, only two species, *Frenkelia glareoli* and *F. microti* have been identified that form cysts in the CNS of rodents. The subfamily Sarcocystinae was characterised by obligate development in two hosts, with the sexual sporogony stage occurring in the intestine of the definitive hosts. Traditionally, the *Frenkelia* and *Sarcocystis* genera were distinguished based on the localisation and morphology of their asexual stages in the intermediate host. The asexual phase of the life cycle of *Frenkelia* species occurs in the liver of the intermediate host, with only one generation of merozoites, while cysts are exclusively detected in the CNS [14]. Molecular analyses conducted between 1998 and 2000 suggested the synonymisation of these two genera [15,16,17]. Finally, the species *F*. *glareoli* was reclassified as *S*. *glareoli*, *F*. *microti* was renamed *S*. *microti* and a new name *S*. *jaypeedubeyi* was proposed for the former *S*. *microti* [18,19].

It has been observed that *S. microti* demonstrates less specificity for intermediate hosts compared to *S. glareoli*. Cysts of *S. microti* have been detected in the brains of rodents belonging to multiple genera, including *Microtus*, *Apodemus*, *Mesocricetus*, *Rattus*, *Mus*, *Mastomys*, *Cricetus*, and *Chinchilla* [1]. Additionally, it is presumed that *S. microti* can also form cysts in the Norway lemming (*Lemmus lemmus*) [20], muskrat (*Ondatra zibethica*) [21] and North American porcupine (*Erethizon dorsatum*) [22]. Meanwhile, some authors believe that *S. glareoli* primarily utilizes the bank vole (*Clethrionomys glareolus*) as its intermediate host [1]. Nevertheless, reports exist demonstrating the presence of *S. glareoli* in the European water vole (*Arvicola amphibius*), common vole (*Microtus arvalis*), short-tailed vole (*M. agrestis*) and members of the genus *Apodemus* [23,24]. Furthermore, based on short fragment of the conservative *18S* rRNA gene, an organism similar to *S*. *glareoli* has been reported in the broad-eared bat (*Nyctinomops laticaudatus*) [25].

*Sarcocystis microti* infections have been documented in small mammals across North America [26], Europe [23,27,28,29], and Japan [30]. Similarly, the presence of *S. glareoli* has been reported in Europe [23,24,28,29,31,32,33]. Both *S. microti* and *S. glareoli* utilize the common buzzard (*Buteo buteo*) as their definitive host. However, the rough-legged buzzard (*Buteo lagopus*) is also suspected to serve as a definitive host for *S. glareoli* [34], while *S. microti* employs red-tailed hawks (*Buteo jamaicensis*) [26]. Thus, the available data do not resolve the issue of intermediate and definitive host specificity of *S*. *glareoli* and *S*. *microti*. Since the identification of *S*. *glareoli* and *S*. *microti* has primarily relied on morphological studies, further molecular investigations are necessary to identify these *Sarcocystis* species in different hosts.

Both *S*. *glareoli* and *S*. *microti* were previously confirmed in Lithuania through microscopic analysis of 560 rodents caught across different regions of the country between 1995 and 2001 [28]. However, *Buteo* buzzards in Lithuania have not been examined for *Sarcocystis* parasites. The objectives of our present study are threefold: (i) to establish the prevalence of *Sarcocystis* spp. in the brains of small mammals collected in Lithuania, (ii) to identify *Sarcocystis* species in the brain tissues of these animals using molecular methods, and (iii) to investigate the potential role of *Buteo* buzzards in the transmission of *Sarcocystis* spp. forming cysts in the brains of small mammals using molecular analysis.

## 2. Materials and Methods

### 2.1. Trapping of Small Mammals

A standard snap trap line method (25 traps at 5 m intervals) was deployed to capture small mammals at 13 distinct locations within Lithuania during the autumn of the year 2023. In three cases, sites in close proximity were combined into one sample (Figure 1). Subsequently, the study analysed 694 individual small mammals representing 11 species: the striped field mouse (*Apodemus agrarius*), *n* = 5, the yellow-necked mouse (*A. flavicollis*), *n* = 2, *C. glareolus, n* = 374, *M. agrestis, n* = 14, *M. arvalis, n* = 144 the root vole (*Alexandromys oeconomus*), *n* = 11, the harvest mouse (*Micromys minutus*), *n* = 3, the house mouse (*Mus musculus*), *n* = 33, the water shrew (*Neomys fodiens*), *n* = 9, the common shrew (*Sorex araneus*), *n* = 45 and the pygmy shrew (*S. minutus*), *n* = 54. The first eight species belong to the order Rodentia, while the last three species are members of the order Eulipotyphla.

### 2.2. Morphological Examination of Sarcocystis spp. from Brain Tissues of Small Mammals

To determine the prevalence of *Sarcocystis* spp. in the brain of small mammals, fragments of brain tissue (~0.2 g) were stained with 0.2% methylene blue solution for 24 h. Then samples were cleared in 1.5% acetic acid solution for 45–50 min. Afterwards, samples were squeezed between glass compressors and studied under a light microscope (LM) at 40× and 100× magnification. The parasite load was evaluated by counting cysts in ~0.2 g of sample. Eventually, the cysts detected in brain samples were morphologically characterised in squashed fresh preparations under LM. Overall, 10 cysts were excreted from the brain from five individual bank voles (isolates CgLt189.1rp; CgLt189.2rp; CgLt388.1rp; CgLt388.2rp; CgLt.782.1rp; CgLt782.2rp; CgLt.963.1rp; CgLt963.2rp; CgLt1124.1rp; CgLt1124.2rp). Genomic DNA was extracted from the isolated individual cysts without delay.

### 2.3. Genetic Characterization of Sarcocystis spp. Detected in Small Mammals

The DNA isolation from sarcocysts was conducted using the GeneJET Genomic DNA Purification Kit (Thermo Fisher Scientific Baltics, Vilnius, Lithuania) according to the manufacturer’s recommendations. The partial *28S* rRNA was amplified using KL-P1F/KL-P1R primer pairs (Table 1) [35]. Each PCR reaction was carried out in a 25 µL mixture containing 12.5 µL of DreamTaq PCR Master Mix (Thermo Fisher Scientific Baltics, Vilnius, Lithuania), 4 µL of DNA template, 0.5 µM of both forward and reverse primers and nuclease-free water. The PCR was initiated with the initial hot start at 95 °C for 5 min followed by 35 cycles of 94 °C for 45 s, annealing at 52 °C for 60 s and 72 °C for 80 s, and a final extension at 72 °C for 7 min. The visualisation, purification, and sequencing of PCR products followed the previously described protocol [36]. In order to detect essentially similar DNA sequences, and evaluate the interspecific and intraspecific genetic variability of detected *Sarcocystis* parasites, the *28S* rRNA sequences generated in this study were compared with those of various *Sarcocystis* spp. using the nucleotide BLAST program (http://blast.ncbi.nlm.nih.gov/, accessed on 10 March 2024).

### 2.4. Collection of Buteo Buzzards

A total of 12 birds (10 *B. buteo* and two *B. lagopus*) were collected between 2017 and 2020. All birds, obtained from the Kaunas Tadas Ivanauskas Zoology Museum (the Lithuanian national authority responsible for monitoring dead birds), were found dead as a result of collisions with motor vehicles, power lines, buildings, etc., and were kept frozen at −20 °C until they were dissected.

### 2.5. Morphological Examination of Sarcocystis spp. from Intestines of Buteo Buzzards

*Sarcocystis* spp. oocysts/sporocysts were excreted from the entire intestine of each buzzard using a slightly modified method by Verma et al. [38]. Initially, the small intestine was removed from the bird. Then, the faeces from each intestine were squeezed, and the gut was cut lengthwise. The intestinal epithelium was scraped with the help of a scalpel and suspended in 50–100 mL of distilled water. The homogenization was performed in a commercial blender at top speed for 1–2 min with breaks to prevent frothing. The homogenate was centrifuged for 6 min at 1600 rpm, 25 °C in a 50 mL centrifuge tube. The supernatant was discarded, and the sediments were re-suspended in 50 mL water. The above-described process was repeated 5–8 times until most oocysts/sporocysts were released from the host tissue. Thereafter, the sporocyst pellet was suspended in HBSS and filtered through cheesecloth. The homogenate was emulsified in 5.25% sodium hypochlorite (bleach) solution (1:1 ratio) in a cold bath for 30 min. The centrifugation for 6 min at 1600 rpm, 25 °C, and removal of supernatant were repeated until the smell of bleach (chlorine) was gone. Eventually, the oocysts/sporocysts of *Sarcocystis* spp. were examined under LM at 400× and 1000× magnification. The 400 μL of re-suspended sediments were taken from each sample and used for DNA extraction.

### 2.6. The Genetic Identification of Sarcocystis spp. from Intestines of Buteo Buzzards

The DNA isolation from the mucosal suspension was performed using the GeneJET Genomic DNA Purification Kit (Thermo Fisher Scientific Baltics, Vilnius, Lithuania). The DNA samples were then stored frozen at −20 °C until further molecular analysis.

*Sarcocystis* spp. were identified by nested PCR of partial *28S* rDNA sequences. In the first step of nested PCR, forward Sgrau281 and reverse Sgrau282 primers were used (Table 1). Whereas in the second step of nested PCR, two primer pairs, GsSglaF1/GsSglaR1 and GsSmicF1/GsSmicR1 were used. The primers of the second round of nested PCR were designed with the help of Primer3Plus program [39]. These primers were designed for amplification of *S*. *glareoli* and *S*. *microti*. However, due to relatively small variance of *Sarcocystis* spp. using rodents and birds as their intermediate and definitive hosts, respectively, within rRNA genes [40], it was suspected that GsSglaF1/GsSglaR1 and GsSmicF1/GsSmicR1 might not only be suitable for the amplification of *S*. *glareoli* and *S*. *microti*. Positive controls (DNA of *S*. *glareoli* extracted from single sarcocysts) were used in each set of PCRs. Three negative controls (nuclease free water instead of target DNA) were used: one control for the first amplification step and two controls for the second step of nested PCR. The third negative control was obtained by transferring 1 µL from the negative control of the first amplification step to the negative control of the second amplification step. PCRs were conducted under the conditions described above in Section 2.3 and using annealing temperatures provided in Table 1. For the second PCR assay, 1 μL from the first PCR assay was used.

The visualisation, purification, and sequencing of PCR products were carried out using the previously described protocol [36]. Our *28S* rRNA sequences were compared with those of various *Sarcocystis* spp. with the Nucleotide BLAST program (megablast option) (http://blast.ncbi.nlm.nih.gov/, accessed on 10 March 2024). The *28S* rDNA sequences of *Sarcocystis* spp. isolated from intestinal scrapings of *Buteo* buzzards were made available in GenBank with PP535696–PP535702 accession numbers.

### 2.7. Phylogenetic Analysis

Based on the molecular results, a putative new *Sarcocystis* species was identified in single sample of intestinal mucosa of *B. lagopus*, prompting further phylogenetic analysis. Phylogenetic relationships were reconstructed using the Maximum Likelihood (ML) method. Multiple alignments of partial *28S* rRNA sequences of 20 *Sarcocystis* taxa was carried out using the MUSCLE algorithm incorporated into MEGA7 version 7.0.26 software [41]. The selection of the evolutionary model was conducted using MEGA7, based on the lowest Bayesian Information Criterion values generated. The robustness of the resulted phylogeny was tested using bootstrap test with 1000 replications.

Furthermore, phylogenetic analysis was performed to evaluate the relationships between the species identified in this study and closely related ones (*S*. *glareoli*, *S*. *microti*, *S*. *jamaicensis* and *Sarcocystis* sp.). A phylogenetic tree was constructed using the Maximum Likelihood algorithm, as described in the paragraph above. Additionally, phylogenetic relationships among the different sequences were inferred through coalescent simulations using a median-joining model implemented in NETWORK v. 10.2.0.0 [42].

### 2.8. Statistical Tests

Confidence intervals (95% CI) for the proportions of infection prevalence were estimated using online software [43]. Differences in infection prevalence between study sites were tested using the G-test on an online calculator [44]. The number of cysts in the samples was compared between the different study sites using the Student’s *t*-test in STATISTICA for Windows, version 6.0 (StatSoft, Inc., Tulsa, OK, USA). The level of significance was set at *p* < 0.05.

## 3. Results

### 3.1. Prevalence and Parasite Load of Sarcocystis spp. in Small Mammals

Cysts of *Sarcocystis* were observed exclusively in the brains of a single small mammal species, *C. glareolus*. The prevalence of infection reached 9.1% (34 infected of 374 tested, 95% CI = 6.4–12.5%). The infection rates varied among these animals across different study areas. *Sarcocystis* spp. were detected in rodents collected from eight out of the 13 localities, accounting for 61.5% of the sampled sites (Table 2). The highest detection rates of cysts of *Sarcocystis* were found in Kamasta, Zabarauskai, and Aukštikalniai. In other sampling sites the prevalence of *Sarcocystis* spp. ranged from 2.1% to 12.5%. The prevalence of *Sarcocystis* spp. infection was significantly different only between the two study sites, Kamasta and Lukštas (G = 7.4, *p* < 0.05).

The number of cysts in 0.2 g brain samples varied from 1 to 50, with an average of 13.4 ± 1.89 (average number of cysts ± standard error). We compared cyst counts across various locations, focusing on three areas where more than one infected individual was captured. The Kamasta study site exhibited the highest average number of cysts (18.0 ± 4.90), correlating with the highest prevalence of examined parasites. The average number of cysts in Kamasta was higher than in the Bileišiai site (11.3 ± 2.24) and the Utena site (9.6 ± 3.17), though the differences were not statistically significant (*t* = 1.45, *p* = 0.16 and *t* = 1.19, *p* = 0.26).

### 3.2. Morphological and Molecular Characterisation of Sarcocystis spp. in Brain Tissues of Small Mammals

The detailed morphological characterisation of *Sarcocystis* parasites observed were carried out in five infected *C. glareolus*. In fresh preparations, the detected cysts greatly varied in shape and size (Figure 2). Sarcocysts were microscopic, appearing as round structures measuring 102.7–279.8 × 102.1–332.8 µm (average size: 189.2 × 207.3 µm; *n* = 4) (Figure 2a), oval structures measuring 38.9–74.0 × 71.5–135.0 µm (average size: 63.3 × 102.7 µm; *n* = 4) (Figure 2b), or irregular round structures measuring 48.2–89.0 × 75.7–104.1 µm (average size: 73.6 × 94.0 µm; *n* = 4) (Figure 2c). Based on molecular analysis, sarcocysts isolated from brain tissues of *C. glareolus* in Lithuania were identified as *S*. *glareoli*.

Our analysis of ten 944 bp long *28S* rRNA sequences of *S*. *glareoli* isolated from *C. glareolus* demonstrated 100% identity among them. These sequences have been deposited in GenBank under the accession number PP535695. Despite significant variation in the size and shape of the cysts, the *28S* rRNA region tested showed no differences among the *S*. *glareoli* isolates.

The 944 bp sequences of *S*. *glareoli* obtained in this study exhibited 99.9% identity to *S*. *glareoli* sequence AF044251. In our sequences, the C insertion was noticed in 537 nucleotide position to be compared (in TAGGT**C**CCCCG region, insertion is bolded and underlined). Notably, numerous other related *Sarcocystis* spp. had the C insertion in a homologous nucleotide position indicating highly probable sequencing error in AF044251 GenBank record.

Additionally, our *28S* rRNA sequences of *S*. *glareoli* differed from those of *S*. *jamaicensis* (KY994650) by two single nucleotide polymorphisms (SNPs), and displayed 98.7% similarity with *S*. *microti* (AF044252), and 98.2% similarity with *S*. *wobeseri* (LR884239), which forms sarcocysts in muscles of birds [35]. Notably, *B. jamaicensis* serves as the definitive host and knockout mouse was shown to be laboratory host of *S*. *jamaicensis* [40,45].

### 3.3. Morphological Characterisation of Sarcocystis spp. Sporocysts/Oocysts Found in Intestines of Buteo Buzzards

Both examined *B. lagopus* harboured sporocysts and sporulated oocysts of *Sarcocystis* spp. (2/2, 100%). Meanwhile, *Sarcocystis* spp. sporocysts and/or sporulated oocysts were detected in eight of the ten investigated *B. buteo* (80%). No oocysts were detected in either host species tested.

Upon microscopical analysis, the free sporocysts of *Sarcocystis* spp. in mucosal scrapings of *B. lagopus* measured 12.6 × 8.4 μm (range: 10.7–15.2 × 5.5–9.6 μm; *n* = 55) (Figure 3a), while sporulated oocysts, each containing two sporocysts, measured 12.3 × 18.8 μm (range: 10.6–13.4 × 18.1–19.5; *n*= 15) in size (Figure 3b). Sporocysts of *Sarcocystis* spp. in *B. buteo* measured 13.0 × 8.9 μm (range: 10.1–15.9 × 7.1–11.0; *n* = 118) (Figure 3c), while sporulated oocysts were 13.0 × 19.0 (range: 11.1–16.3 × 16.1–25.2; *n* = 59) (Figure 3d).

The examination of sporocysts and sporulated oocysts found in the intestines of *B. lagopus* and *B. buteo* indicated that morphological parameters of *Sarcocystis* spp. overlapped in size.

### 3.4. Molecular Investigation of Sarcocystis spp. Detected in Intestines of Buteo Hawks

When analysing samples of DNA extracted from scrapings of the intestinal mucosa of two hawk species, amplification was unsuccessful using GsSmicF1 and GsSmicR1 primers in the second round of nested PCR. Out of the 10 examined intestinal mucosa samples from *B. buteo*, five tested positive (50.0%) by nested PCR using GsSglaF1 and GsSglaR1 as internal primers. The resulting five sequences of *28S* rRNA without primer binding sites were 515 bp long. These sequences showed 100% identity with each other and to those of *S*. *glareoli* from *C. glareolus* from Lithuania.

Based on BLAST analysis, the *28S* rRNA sequences obtained in this study differed from those of *S*. *jamaicensis* (KY994650) by two T/G and A/G SNPs (99.6% similarity), and from those of *S*. *microti* (AF044252) by four SNPs and single indel (99.0% similarity), and by ≥2.3% from those of other *Sarcocystis* spp. available in GenBank. Therefore, molecular investigation confirmed the presence of *S*. *glareoli* in the intestinal mucosa of five common buzzards.

Sequencing of the *28S* rRNA fragment amplified by the GsSglaF1/GsSglaR1 primer pair, both *B. lagopus* were positive for *Sarcocystis* parasites. Comparison of the analysed 515 bp long and 522 bp long sequences revealed the presence of two different *Sarcocystis* species, *Sarcocystis* sp. Rod3 and *Sarcocystis* sp. Rod4.

The BLAST comparison showed that the 515 bp long sequence of *Sarcocystis* sp. Rod3 differed from those of *S*. *jamaicensis* (KY994650), *S*. *glareoli* (AF044251) and *S*. *microti* (AF044252), by one C/G SNP, by two SNPs and single indel, and by three SNPs and single indel, respectively. Based on investigated *28S* rRNA fragment, *Sarcocystis* sp. Rod4 notably differed from *Sarcocystis* spp. forming cysts in the brains of rodents (94.1–94.6% similarity). The 522 bp long sequence of *Sarcocystis* sp. Rod4 displayed 96.4% similarity with that of *S*. cf. *strixi* (OQ557459) from *A. flavicollis*, 96.2% similarity with that of *S*. *funereus* (MW349707), and 95.8% similarity with that of *S*. *strixi* (MF162316).

It is worth noting that the experimental intermediate host of *S*. *funereus* and *S*. *strixi* are laboratory mouse, while owls serve as definitive hosts of these *Sarcocystis* species [7,46,47].

Our 515 bp long *28S* rRNA sequences of *S*. *glareoli* and *Sarcocystis* sp. Rod3 exhibited significant similarity to 299–316 bp long sequences (OK576420, OK576425, OK576428-36, OK576438-43, OK576447-49, OK576451, OK576456, OK576459, OK576461-62) of *Sarcocystis* parasite obtained from intestinal samples of two hosts, *B. jamaicensis* and red-shouldered hawks (*Buteo lineatus*) from the USA. This *Sarcocystis* parasite, as identified by Rogers et al. (2022), was classified as *S*. *jamaicensis* [48]. However, in this study, we refer to it as *Sarcocystis* sp. Based on *28S* rRNA sequences, *S*. *glareoli* and *Sarcocystis* sp. Rod3 shared 97.0–99.7% and 97.0–100% similarity with *Sarcocystis* sp. isolated from *B. jamaicensis*, and *B. lineatus*.

### 3.5. Phylogenetic Relationships of Examined Sarcocystis Species

Based on the *28S* rRNA fragment amplified with GsSglaF1/GsSglaR1 primers, the three *Sarcocystis* species identified in this study were categorized into two clusters, alongside *Sarcocystis* species utilizing rodents as intermediate hosts and birds of prey as definitive hosts (Figure 4). Our sequences of *S*. *glareoli* grouped together with the sequence of *S*. *glareoli* taken from the GenBank AF044251. *Sarcocystis* sp. Rod3 was found to be most closely related to *S*. *jamaicensis* and *S*. *microti*. However, the grouping of *Sarcocystis* sp. Rod3 with *S*. *microti* was not well supported (68 bootstrap value). Notably, the branch lengths within the clade comprising *S*. *glareoli*, *S*. *jamaicensis*, *Sarcocystis* sp. Rod3 and *S*. *microti* was shorter compared to that established in another clade consisting of *Sarcocystis* spp. using rodents and birds of prey as their intermediate and final hosts. The following clade included *S*. *funereus*, *S*. *strixi*, *S*. cf. *strixi* and *Sarcocystis* sp. Rod4 from *B. lagopus*. *Sarcocystis funereus* formed a sister branch to *S*. *strixi* and *S*. cf. *strixi*, while *Sarcocystis* sp. Rod4 formed a sister branch to the aforementioned taxa.

We conducted another phylogenetic analysis by comparing our *S*. *glareoli* and *Sarcocystis* sp. rod3 *28S* rRNA sequences with those of *Sarcocystis* sp. from Rogers et al. [48] as well as those of *S*. *glareoli* (AF044251), *S*. *microti* (AF044252) and *S*. *jamaicensis* (KY994650). Out of the 32 analysed sequences, 13 haplotypes were identified, and four haplotypes (sequences: OK576420, OK576425, OK576431, OK576432, OK576433, OK576435, OK576436, OK576440, OK576447, OK576451, OK576459, OK576461, OK576462, OK576464, OK576465) were clustered together with a high support value of 89 (Figure 5a). These four haplotypes were separated from the remaining ones by at least six mutations (Figure 5b) and presumably belonged to a single *Sarcocystis* species isolated from *B. jamaicensis* and *B. lineatus*. However, clustering of other haplotypes belonging to *S*. *glareoli* and *Sarcocystis* sp. rod3, *S*. *jamaicensis*, *S*. *microti* and remaining haplotypes of *Sarcocystis* sp. (OK576428, OK576429, OK576430, OK576434, OK576438, OK576439, OK576441, OK576442, OK576443, OK576448, OK576449, OK576456) was not well defined. Thus, the variability of the examined fragment was not sufficient to reveal how many distinct taxa comprised the nine haplotypes. It also should be pointed out that at 307–308 bp *28S* rRNA fragment, *Sarcocystis* sp. rod3 showed 100% identity to OK576428 and OK576430 sequences of *Sarcocystis* sp. from two *Buteo* buzzard species from the USA.

## 4. Discussion

### 4.1. Sarcocystis Parasites in the Brain Tissues of Rodents

A previous study conducted in Lithuania reported that 72 out of 341 (21.1%) of the analysed *C. glareolus* had cysts of *S. glareoli* in their brains [28]. However, in the present examination, the prevalence of *S. glareoli* was significantly lower (*p* < 0.001), as only 9.1% (95% CI = 6.4–12.5) of the studied *C. glareolus* specimens were infected.

According to published data from other European countries, the infection rates of *S*. *glareoli* in the *C. glareolus* ranged from 1.0% to 47.3% [23,24,29,31,32,33]. The prevalence of *S. glareoli* identified in our study is consistent with data from one study in Germany, which reported that 10.3% (45/445) of studied *C. glareolus* were positive for *S*. *glareoli* [33]. The highest infection rates of *S. glareoli* in *C. glareolus* were reported in France (47.3%, 178/376) [23] and the Czech Republic [29], followed by a study in Germany (22.6%, 59/257) [24]. Meanwhile, the lowest prevalence was established in another study conducted in France (1.0%, 1/98) [32] and in a study from the UK (6.3%, 1/16) [31]. Thus, the detection rates of this parasite in *C. glareolus* vary considerably even within the same countries.

In this study, no cysts of *S. microti* were detected, despite its previous detection in Lithuania. Formerly, *S. microti* was identified in two rodent species; however, the prevalence of this parasite was low, with rates of 2.9% (1/35) in *A. oeconomus* and 4.2% (1/24) in *M. agrestis* [28]. Two comprehensive investigations (*n* > 1700) carried out in Europe determined a lower prevalence of *S*. *microti* compared to *S*. *glareoli* [23,29]. In the study conducted in France, cysts of *S. microti* were found in *M. agrestis*, *M. arvalis*, and *C. glareolus*, with prevalence ranging from 1.0% to 9.2% in each species [23]. Meanwhile, in the Czech Republic, *S. microti* was detected in *M. arvalis*, *C. glareolus*, and *Apodemus* sp., with infection rates ranging from 0.6% to 5.0% in all tested species [29]. Notably, *S*. *microti* was detected in 20.7% (6/29) of Bedford’s red-backed voles (*Clethrionomys rufocanus bedfordiae*) in Japan [30].

There are multiple reports that the prevalence of *Sarcocystis* spp. in small mammals is affected by seasonal changes. A study on Norwegian lemmings (*Lemmus lemmus*) concluded that *S. microti* infestation of intermediate hosts depends on the season, with the highest number of parasites found in summer (9%) and the lowest in autumn (0–4%) [27]. Furthermore, research conducted in the Netherlands analysed seasonal dynamics of *S. cernae* infection in the intermediate host *M. arvalis* and definitive host the common kestrel (*Falco tinnunculus*) and concluded that prevalence of *S. cernae* varies throughout the seasons and was highest in the spring [49]. Similar results were obtained in a study in France, where a variety of small mammals belonging to genera *Myodes*, *Arvicola*, *Microtus*, *Apodemus* and *Sorex* were investigated [23]. In our study, rodents were collected in autumn when lower infection rates of *Sarcocystis* spp. are observed based on aforementioned research. In the previous study conducted in Lithuania [28] rodents were collected not only in autumn, but also in spring and summer thus probably leading to the detection of higher total prevalence of *Sarcocystis* spp. in the brains of rodents.

### 4.2. Buteo Buzzards as Definitive Hosts of Sarcocystis spp.

Although detailed studies on *S*. *glareoli* and *S*. *microti* have been carried out in intermediate hosts [23,28,29], these parasites have been scarcely studied in their natural definitive hosts. Based on morphological examination, “*Frenkelia*-like” sporocysts were detected in the droppings and intestines of *Buteo* buzzards in Europe [29,50] and North America [26,51,52]. Without molecular analysis, it is not possible to ascertain whether the sporocysts discovered in the above studies belonged to *S*. *glareoli* and/or *S*. *microti*, which form cysts in the brains of rodents, or to some other *Sarcocystis* spp. This is because the morphometrical parameters of sporocysts of many different species of *Sarcocystis* overlap [1,52].

Due to the lack of data, the prevalence of *Sarcocystis* spp. sporocysts in different *Buteo* species, geographical regions, seasons, etc. is still generally unknown. A detailed study conducted in the Czech Republic revealed that the prevalence of *Sarcocystis* spp. in faeces from nests of *B. buteo* ranged from 28.3% to 88.6%, depending on the season and locality [29]. Another study performed in North America examined droppings from red-tailed hawks (*Buteo borealis*) in a rehabilitation centre and determined that half of them contained *Sarcocystis* spp. sporocysts [52]. In this work, by microscopical examination, 80% of *B. buteo* and both analysed *B. lagopus* were infected with oocysts/sporocysts of *Sarcocystis* spp. Based on *28S* rRNA sequence analysis, *S*. *glareoli* was confirmed in *B. buteo* and two *Sarcocystis* species (*Sarcocystis* sp. Rod3 and *Sarcocystis* sp. Rod4) were detected in *B. lagopus*. Thus, our results indicate that *Buteo* buzzards in Europe can transmit not only *S*. *glareoli* and *S*. *microti*, previously assigned to the genus *Frenkelia*.

The common buzzard nests across nearly the entire European region within the Western Palearctic [53]. This migratory species breeds in Lithuania and is commonly found throughout the country, although it rarely stays for the winter season. It nests in woodlands and small clusters of trees near cultivated fields, displaying remarkable adaptability in choosing its breeding sites [54]. A wide array of prey is consumed, encompassing small mammals, birds, reptiles, amphibians, larger insects, and earthworms [55]. The primary portion (38.3%) of *B. buteo* diet consists of small mammals, predominantly voles [54,56]. Studies conducted in Poland indicate that these raptors mostly catch common voles [57]. Of the five *B. buteo* in which *S*. *glareoli* were identified, only two of them have a known collection date (February and September). Taking into account that definitive hosts excrete sporocysts of *Sarcocystis* spp. up to several months [1] and based on migratory data, birds collected in February and September were most likely infected with *S*. *glareoli* in Lithuania.

Another species, *B. lagopus*, only winters in Lithuania. This buzzard is distributed in the northern part of the Western Palearctic. Its range stretches eastward, predominantly across the tundra belt, reaching the Pacific Ocean. From there, it tracks along the coastal areas southward to Sakhalin and extends across the northern reaches of North America on the opposite side of the Pacific [53]. The rough-legged buzzard predominates in the diet, mainly small rodents, accounting typically for 80–95% of the food, where they occur in large numbers [52,54]. In Lithuania, ringed *B. lagopus* chicks were found in central Europe: Slovakia, Hungary, France, and Italy [54]. In autumn and winter in Lithuania, the ringed *B. lagopus* come from Norway, Finland, and mostly from Sweden [54]. Both of *B. lagopus* examined in current work were collected in November of 2018. Based on ecological data, infected *B. lagopus* were migrants most likely from Scandinavia [54]. It is likely that these buzzards were infected with *Sarcocystis* sp. Rod3 and *Sarcocystis* sp. Rod4 before entering Lithuania.

### 4.3. Genetic Identification of Sarcocystis sp. Rod3 and Sarcocystis sp. Rod4

Based on phylogenetic analysis using *28S* rRNA, *Sarcocystis* sp. Rod3 was found to be closely related to three *Sarcocystis* species, *S*. *glareoli*, *S*. *microti* and *S*. *jamaicensis* (Figure 4 and Figure 5). These *Sarcocystis* species produce sarcocysts with thin and smooth cyst walls [1,40]. The natural intermediate host of *S*. *jamaicensis* is unknown. However, laboratory experiments have shown that this species circulates between rodents and *B. jamaicensis* [40,45]. Two IFN-c gene knockout mice orally infected with sporocysts of *S*. *jamaicensis* had meningoencephalitis associated with schizonts and merozoites of this parasite, while sarcocysts were detected in the muscles of the animals [45]. Thus, the developmental stages of these three species, *S*. *glareoli*, *S*. *microti* and *S*. *jamaicensis*, are found in the brain. The detection of *Sarcocystis* sp. Rod3 in the intestine of a single *B. lagopus* collected in Lithuania suggests that it might be another species in Europe, apart from *S*. *glareoli* and *S*. *microti*, which can infect rodent brains.

Also, our work demonstrated that *28S* rRNA locus is not sufficiently variable for the reliable discrimination of *S*. *glareoli*, *S*. *microti*, *S*. *jamaicensis* and *Sarcocystis* sp. Rod3. To date, only sequences of *18S* rRNA and *28S* rRNA of *S*. *glareoli* and *S*. *microti* are available in GenBank [15,16], while *S*. *jamaicensis* has been characterised within *18S* rRNA, 28S rRNA, mitochondrial cytochrome oxidase 1 (*cox1*) and *ITS1*. The *ITS1* genetic marker has been proven to be the best choice for the separation of *Sarcocystis* species using birds and predatory mammals as their intermediate hosts [58,59]. Although this genetic region has not been commonly applied for the characterization of *Sarcocystis* spp. from rodents, it has shown more variability compared to *cox1* [9,13,59]. In contrast, *cox1* is relatively conservative considering *Sarcocystis* spp. parasitizing small mammals [59]. Therefore, the highly variable *ITS1* can be helpful for the distinguishing the aforementioned *Sarcocystis* species.

Furthermore, the results of the current study and that described by Rogers et al. [48] indicate that different species of the genus *Buteo* from North America and Europe can transmit the same *Sarcocytis* species. Intriguingly, based on short at 307–308 bp *28S* rRNA sequences, *Sarcocystis* sp. Rod3 was identified in three different *Buteo* species. Nevertheless, further studies are needed to clarify the specificity of *Sarcocystis* species for *Buteo* buzzards as their definitive hosts.

*Sarcocystis* sp. Rod 4, identified in a single *B. lagopus*, phylogenetically clustered with *S*. *funereus*, *S*. *strixi*, and *S*. cf. *strixi* (Figure 4). Two owl species, the Tengmalm’s owl (*Aegolius funereus*) and the barred owl (*Strix varia*), are proven definitive hosts of *S*. *funereus* [47] and *S*. *strixi* [46], respectively. Sarcocysts of these two *Sarcocystis* species have not yet been discovered in natural intermediate hosts. However, recently the DNA of *S*. *funereus* has been detected in blood sample of a single *C. glareolus* from Lithuania [60]. There is a lack of investigations on the role of the Strigiformes in the transmission of *Sarcocystis* spp. [37,46,47]. Even so, the close relationships of *Sarcocystis* sp. Rod 4 to *S*. *funereus* and *S*. *strixi* imply that *Sarcocystis* species using Accipitriformes and Strigiformes as their definitive hosts might not have evolved separately. Further examinations are needed to clarify whether the same *Sarcocystis* species can employ representatives of both orders Accipitriformes and Strigiformes as their definitive hosts.

## 5. Conclusions

After examining brain samples from 694 small mammals belonging to eight Rodentia and three Eulipotyphla species, cysts of *S*. *glareoli* were observed in only 9.1% of the *C. glareolus* investigated. The literature suggests that this molecularly confirmed *Sarcocystis* species is more common in Europe compared to another species, *S*. *microti*, which also forms cysts in the brains of small mammals. The *28S* rRNA sequence analysis confirmed the presence of *S*. *glareoli* in half of the *B. buteo* examined, indicating that these predatory birds are natural definitive hosts of *S*. *glareoli* in Lithuania. Our results also highlight the necessity of using other genetic markers, such as *ITS1*, to distinguish between *Sarcocystis* spp. parasitizing the brains of small mammals.

Based on molecular examination, two species were identified in *B. lagopus*: *Sarcocystis* sp. Rod3, which is closely related to *S*. *jamaicensis*, *S*. *glareoli* and *S*. *microti*, and a putative new species, *Sarcocystis* sp. Rod4. Ecological data suggest that *B. lagopus* were likely infected outside Lithuania. Overall, our work underscores the significant role of *Buteo* buzzards in the transmission of *Sarcocystis* parasites that produce cysts in small mammals.

## Figures and Tables

**Figure 1 biology-13-00264-f001:**
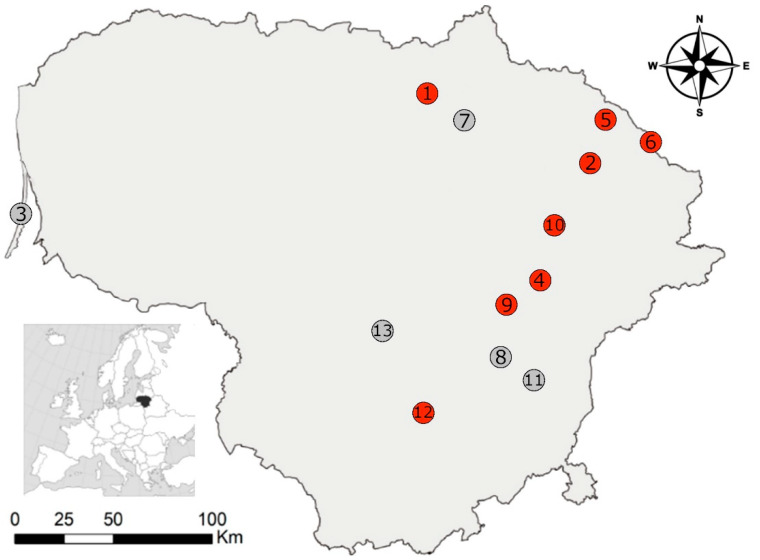
Investigation sites in Lithuania where brain tissues of small mammals were examined for the presence of *Sarcocystis* pathogens: 1: Aukštikalniai; 2: Bileišiai; 3: Juodkrantė; 4: Kamasta; 5: Kukinis; 6: Lukštas (including also Stelmužė locality); 7: Mieliūnai (including Deikiškiai locality); 8: Sudervė (including Brinkiškės and Saldenė localities); 9: Šešuolėliai; 10: Utena; 11: Vilnius; 12: Zabarauskai; and 13: Žiežmariai. Red dots indicate *Sarcocystis* spp. detected; grey dots indicate not detected.

**Figure 2 biology-13-00264-f002:**
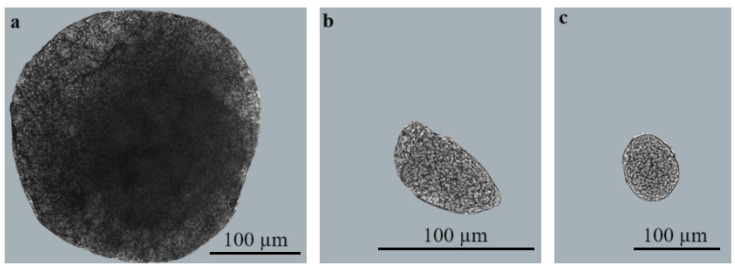
Sarcocysts in brain tissues of *C. glareolus*: (**a**): round form, (**b**): oval form, (**c**): irregular round form.

**Figure 3 biology-13-00264-f003:**
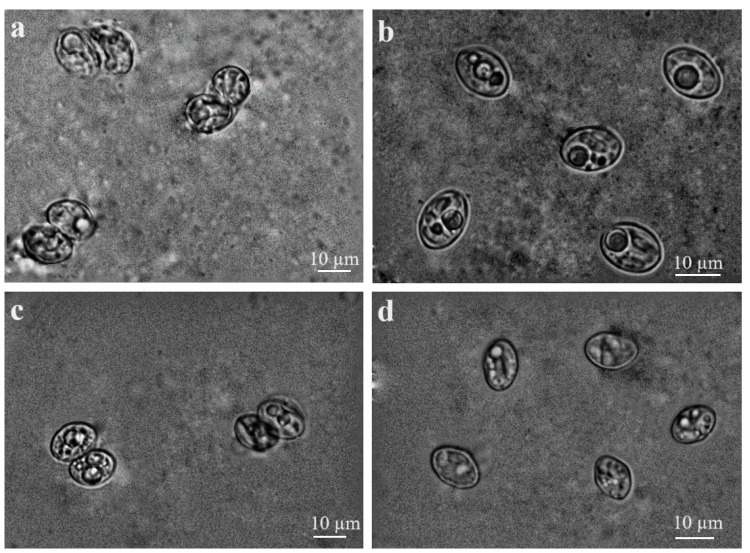
Sporulated oocysts/sporocysts of *Sarcocystis* spp. found in small intestine mucosal scrapings of buzzards: (**a**,**c**): sporulated oocysts; (**b**,**d**): sporocysts; (**a**,**b**): *Sarcocystis* spp. from *B. lagopus*; (**c**,**d**): *Sarcocystis* spp. from *B. buteo*.

**Figure 4 biology-13-00264-f004:**
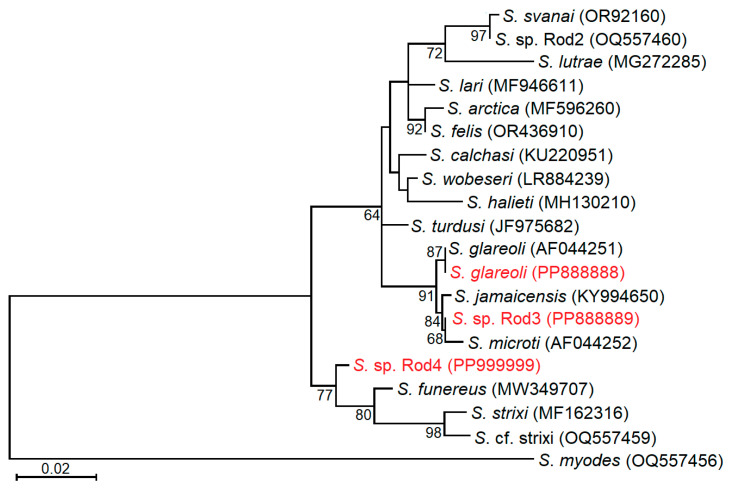
Maximum likelihood phylogenetic tree of *Sarcocystis* spp. based on *28S* rRNA alignment containing 532 nucleotide positions. *Sarcocystis myodes* was chosen as an outgroup. The HKY + G evolutionary model was used for analysis. GenBank accession numbers of sequences are displayed in parenthesis. Bootstrap values higher than 50 are indicated next to branches. Three *Sarcocystis* species identified in intestinal mucosa of buzzards from Lithuania are highlighted in red.

**Figure 5 biology-13-00264-f005:**
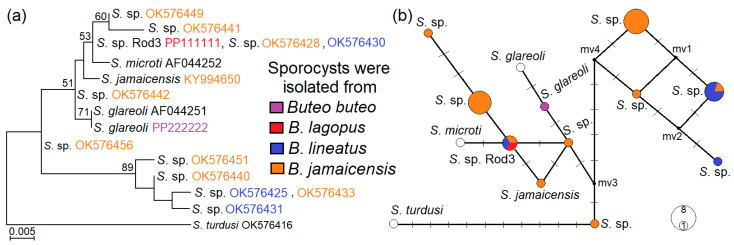
The maximum likelihood phylogram (**a**) and median-joining network (**b**) of selected *Sarcocystis* spp. based on 299 bp long fragment of *28S* rRNA. *Sarcocystis turdusi* was used as an outgroup. The Tamura-Nei + G + I evolutionary model was used for ML analysis. GenBank accession numbers are given after species name. The figures next to branches display bootstrap values higher than 50. When multiple sequences were assigned to a single haplotype, only sequences from different hosts are shown in the figure. Hypothetical not determined haplotypes are named mv1–mv4. Dashes indicate mutational steps. Colours respond to *Buteo* buzzards from which *Sarcocystis* spp. oocysts/sporocysts were isolated. The *S*. sp. in our figure correspond to *S*. *jamaicensis* in [48].

**Table 1 biology-13-00264-t001:** List of oligonucleotides used for the identification of *Sarcocystis* spp.

Primer Name	Orientation	Primer Sequence	Type of PCR	T_a_	LP	Reference
KL-P1F	Forward	TACCCGCTGAACTTAAGCAT	conventional	52	~1000	[35]
KL-P1R	Reverse	CCCAAGTTTGACGAACGATT	conventional			
Sgrau281	Forward	GAACAGGGAAGAGCTCAAAGTG	nested	63	~900	[37]
Sgrau282	Reverse	GGTTTCCCCTGACTTCATTCTAC	nested			
GsSglaF1	Forward	GCAAAATGTGTGGTAAGTTTCACAT	nested	61	~560	Present study
GsSglaR1	Reverse	CCCTCTAAAAAGATGTTACCCTTCT	nested			
GsSmicF1	Forward	TGTGGTAAGTTTCACATAAGGCTAA	nested	61	~550	
GsSmicR1	Reverse	CTTTCTAAAAAGATGTACCTTCTCCT	nested			

T_a_: annealing temperature expressed in °C, LP: length of the product in bp.

**Table 2 biology-13-00264-t002:** The quantity of examined specimens of small mammals across 13 study sites, with average detection rates per site and the count of *Sarcocystis*-infected individuals provided in parentheses. Abbreviations: DR: detection rate, DR CI: 95% confidence interval of detection rate, *A. agr*: *Apodemus agrarius*, *A. fla: A. flavicollis*, *C. gla: Clethrionomys glareolus*, *M. agr: Microtus agrestis*, *M. arv: M. arvalis*, *A. oec: A. oeconomus*, *M. min: Micromys minutus*, *M. mus: Mus musculus*, *N. fod: Neomys fodiens*, *S. ara: Sorex araneus*, *S. min: S. minutus*.

Investigation Sites	DR, %	DR CI	Species										
			*A. agr*	*A. fla*	*C. gla*	*M. agr*	*M. arv*	*A. oec*	*M. min*	*M. mus*	*N. fod*	*S. ara*	*S. min*
1. Aukštikalniai	16.7	0.4–64.1			6 (1)		7 (0)						
2. Bileišiai	10.0	5.7–15.9		1 (0)	150 (15)	5 (0)	16 (0)		1 (0)		1 (0)	16 (0)	31 (0)
3. Juodkrantė					9 (0)								
4. Kamasta	22.5	10.8–38.5			40 (9)	1 (0)	3 (0)					10 (0)	8 (0)
5. Kukinis	12.5	0.3–52.6			8 (1)	2 (0)	5 (0)					3 (0)	1 (0)
6. Lukštas	2.1	0.1–11.3		1 (0)	47 (1)	1 (0)	7 (0)	2 (0)			1 (0)	2 (0)	3 (0)
7. Mieliūnai			5 (0)		2 (0)		53 (0)	4 (0)		21 (0)		3 (0)	
8. Sudervė					24 (0)		21 (0)		1 (0)	5 (0)	6 (0)	4 (0)	1 (0)
9. Šešuolėliai	7.7	0.2–36.0			13 (1)								
10. Utena	7.6	2.5–16.8			66 (5)	4 (0)	30 (0)	5 (0)	1 (0)	2 (0)	1 (0)	7 (0)	10 (0)
11. Vilnius										5 (0)			
12. Zabarauskai	20.0	0.5–71.6			5 (1)	1 (0)	1 (0)						
13. Žiežmariai					4 (0)		1 (0)						
Total:			5 (0)	2 (0)	374 (34)	14 (0)	144 (0)	11 (0)	3 (0)	33 (0)	9 (0)	45 (0)	54 (0)

## Data Availability

The 28S rDNA sequences of *Sarcocystis* spp. generated in the present study were submitted to the GenBank database under PP535695–PP535702.

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
