# Peer review of "First Observations of Buzzards (Buteo) as Definitive Hosts of Sarcocystis Parasites Forming Cysts in the Brain Tissues of Rodents in Lithuania"

_biology, 2024, doi:10.3390/biology13040264_

Round 1

Reviewer 1 Report

Comments and Suggestions for Authors

This study presents data on the morphological and molecular characterizations of Sarcosystis species in buzzards and almost 700 rodents in Lithuania. I congratulate authors because bbviously a lot of time and effort has been invested in collection and analysis of the samples.

Overall, the manuscript is well-written though I have some minor comments for its improvement:

- L54: "Some species of Sarcocystis are pathogenic". Do authors mean some species are not pathogenic? 

- L57: correct "Tte" as "The"

- Since birds kept frozen (for years?) and at the time of experiments intestinal tissue and contents were homogenized in a commercial blender at top speed, how recovery of intact and sporulated oocysts/sporocysts could happen?

- L431: type "Frenkelia" in italics

- L477-479: "species were not previously found in small mammals from Lithuania" does not make it a good reason that "these species employ intermediate hosts that are not prevalent in Lithuania". revise

- L480-491 is somehow repeating L464-479.

- L540: delete "a" in front of "half"

- L549: what are "potential adverse effects of these parasites on intermediate hosts."

Author Response

Reviewer 1 comments and answers

Comment: This study presents data on the morphological and molecular characterizations of Sarcosystis species in buzzards and almost 700 rodents in Lithuania. I congratulate authors because bbviously a lot of time and effort has been invested in collection and analysis of the samples.

Overall, the manuscript is well-written though I have some minor comments for its improvement

Answer: thank you for your work and comments.

Comment: L54: "Some species of Sarcocystis are pathogenic". Do authors mean some species are not pathogenic?

Answer: Only some species are known to be pathogenic, a large proportion of Sarcocystis species are asymptomic, and the pathogenicity of a number of species has not been sufficiently investigated. We, therefore, do not change the sentence.

Comment: L57: correct "Tte" as "The"

Answer: done

Comment: Since birds kept frozen (for years?) and at the time of experiments intestinal tissue and contents were homogenized in a commercial blender at top speed, how recovery of intact and sporulated oocysts/sporocysts could happen?

Answer: Storage in a frozen animal did not affect the morphological detection of sporocysts, but such sporocysts are non-viable and cannot be used for infection experiments

Comment: L431: type "Frenkelia" in italics

Answer: done

Comment: L477-479: "species were not previously found in small mammals from Lithuania" does not make it a good reason that "these species employ intermediate hosts that are not prevalent in Lithuania". revise

Answer: Thank you for this remark. We have deleted these sentences.

Comment: L480-491 is somehow repeating L464-479.

Answer: duplicate text deleted, but please note, that this information concerns two different buzzard species.

Comment: L540: delete "a" in front of "half"

Answer: done

Comment: L549: what are "potential adverse effects of these parasites on intermediate hosts."

Answer: We have deleted this sentence, since the potential adverse effects of this parasite is not discussed in the article

Reviewer 2 Report

Comments and Suggestions for Authors

The study by Lithuanian authors deals with intermediate (small wild rodents) and definitive (birds of prey) hosts of Sarcocystis spp. on the territory of Lithuania. The authors' research on Sarcocystis species is of interest in point of view of the wide distribution of these parasites in the world. This study has both theoretical and important practical implications.

The introduction contains an analysis of a sufficient literature review on the research topic. The aims of the study are clear. Clearly presented Materials and Methods, Results and extensive Discussion indicate the high quality of the work carried out by the Authors. The merits of the article include the use of molecular morphological research methods by the authors. The manuscript is well illustrated. I congratulate the Authors for the great efforts made in carrying out a high-quality work.

I have some remarks about this article.

1. The title of the manuscript needs to be rephrased. In this form the meaning is somewhat unclear. Something like this: “First observations (or records ?) of buzzards (Buteo) as definitive hosts of Sarcocystis parasites forming cyst in the brain tissues of rodents in Lithuania”

2. In materials and Methods, it is necessary to indicate how many different small mammals were caught. Therefore, it is better to present Table 2 here (after line 112).

3. Line 102 may be better: Trapping of Small Mammals.

4. The results (lines 223–233) should be presented in a table It will be clearer this way.

5. According International Code of Zoological Nomenclature (ICZN) in scientific articles at the first mention of genus or species its full Latin name with the author and year of description should be given; in relation all species Sarcocistis and their intermediate and definitive hosts (lines 39, 52, 57-60, 73-76, etc.). On subsequent mentions, the generic name is abbreviated to the first letter.

6. In Simple Summary and Abstracts Latin species name must be written in full: Sarcocystis glareoli (line 17), Sarcocystis glareoli and Sarcocystis microti (line 25), Buteo lagopus (line 28).

7. Lines 388–390 – Unnecessary information here. You can delete this.

8. In scientific articles it is better to use the Latin names of organisms. Try to avoid using common animal names in your text. They can be used once at the first mention, and further in the text only Latin. It is better to use the Latin names of buzzards and rodents.

And minor remarks:

Line 57 – The.

Line 197 – remove italics “Buteo buzzards”

The manuscript deserves to be published in Biology, but minor corrections are needed.

Author Response

Reviewer 2 comments and answers

The study by Lithuanian authors deals with intermediate (small wild rodents) and definitive (birds of prey) hosts of Sarcocystis spp. on the territory of Lithuania. The authors' research on Sarcocystis species is of interest in point of view of the wide distribution of these parasites in the world. This study has both theoretical and important practical implications.

The introduction contains an analysis of a sufficient literature review on the research topic. The aims of the study are clear. Clearly presented Materials and Methods, Results and extensive Discussion indicate the high quality of the work carried out by the Authors. The merits of the article include the use of molecular morphological research methods by the authors. The manuscript is well illustrated. I congratulate the Authors for the great efforts made in carrying out a high-quality work.

Answer: thank you

I have some remarks about this article.

Comment: 1. The title of the manuscript needs to be rephrased. In this form the meaning is somewhat unclear. Something like this: “First observations (or records ?) of buzzards (Buteo) as definitive hosts of Sarcocystis parasites forming cyst in the brain tissues of rodents in Lithuania”

Answer: we changed Title as recommended

Comment: 2. In materials and Methods, it is necessary to indicate how many different small mammals were caught. Therefore, it is better to present Table 2 here (after line 112).

Answer: We have provided number of the individuals of every examined species.

Comment: 3. Line 102 may be better: Trapping of Small Mammals.

Answer: acknowledged

Comment: 4. The results (lines 223–233) should be presented in a table It will be clearer this way.

Answer: on your request, detection rates were added to the Table 2. This, however, resulted Table in landscape format.

Comment: 5. According International Code of Zoological Nomenclature (ICZN) in scientific articles at the first mention of genus or species its full Latin name with the author and year of description should be given; in relation all species Sarcocistis and their intermediate and definitive hosts (lines 39, 52, 57-60, 73-76, etc.). On subsequent mentions, the generic name is abbreviated to the first letter.

Answer: we accept comment and included common name and Latin name of species on the first mention. As we do not deal with taxonomy of mammals here, authors were not included. As for the genus name, it is mentioned in a common sense, just to point out that species are of the same genus. We do not analyse any genus as such.

Comment: 6. In Simple Summary and Abstracts Latin species name must be written in full: Sarcocystis glareoli (line 17), Sarcocystis glareoli and Sarcocystis microti (line 25), Buteo lagopus (line 28).

Answer: done

Comment: 7. Lines 388–390 – Unnecessary information here. You can delete this.

Answer: deleted

Comment: 8. In scientific articles it is better to use the Latin names of organisms. Try to avoid using common animal names in your text. They can be used once at the first mention, and further in the text only Latin. It is better to use the Latin names of buzzards and rodents.

Answer: done, with exception of cases where sentence is starting with species name.

And minor remarks:

Comment: Line 57 – The.

Answer: done

Comment: Line 197 – remove italics “Buteo buzzards”

Answer: done

Comment:The manuscript deserves to be published in Biology, but minor corrections are needed.

Answer: thank you